# Leveraging interdependencies among platform and complementors in innovation ecosystem

**Fangcheng Tang[1], Zeqiang Qian[2]** *

1 School of Economics and Management, Beijing University of Chemical Technology, Beijing, China,
2 School of Economics and Management, Beijing University of Chemical Technology, Beijing, China

* 2018400160@mail.buct.edu.cn

**Data Availability Statement:** All relevant data are within the manuscript and its Supporting Information files.

**Funding:** This work was supported by financial support from the National Science Fund of China

## Abstract

In the platform-based innovation ecosystem, the symbiotic evolution and interaction between the participating entities have drawn extensive attention. However, there is a lack of understanding of the co-specialization and co-innovation among members of the innovation ecosystem. This paper addresses this gap. In this paper, the collaborative innovation between a platform firm and different kinds of complementors, and its effects on the performance of the innovation ecosystem are simulated through an improved NK model. The result shows that the co-innovation performance between generalist complementors and the platform firm is generally outperformed than that between the specialist complementors and the platform firm. As the interdependencies between the complementary components get stronger, the innovation performance of the innovation ecosystem influenced by the interaction between different types of complementors and platform firms tend to be an inverted U-shape.

## 1 Introduction

In the digital era, the innovation ecosystem formed by the cross boundary alliances of many participants relying on a platform is transforming the enterprises' strategies [1]. Many of the most influential and valuable companies such as Cisco, Apple, Google and Alibaba have adopted a platform-based business model [2–6]. These companies are promoting the formation and development of their platform-centric innovation ecosystems. Throughout the world, those well-known enterprises, almost without exception, are using innovation ecosystems to produce huge benefits. According to Adner [7], platform-based Innovation Ecosystem (PIE) represents an innovation network that is formed via its platform integrating a large number of innovation resources [7]. The network includes different organizations and stakeholders, which being connected by symbiotic relations will collaborate with and complement each other to innovate. Similar to the definition in biology, the different species are closely related and live in a mutually beneficial style that can promote a positive cycle. Innovation ecosystem is where its participants can constantly promote the development and evolution of the ecosystem by sharing their information and resources, complementing each other with their own

(NSFC) under contract No.71532003, Fundamental Research Funds for the Central Universities under contract No.buctrc201804, Funds for First-class Discipline Construction No. XK1802-5.

**Competing interests:** The authors have declared that no competing interests exist.

products and technologies and taking advantage of the network synergy. PIE signifies a valuable co-production progress where ecological communities will finally break their industry boundaries and integrate and accumulate the advantageous and complementary resources for the innovation process.

Previous research suggests the main participants of PIE are platform firm and complementors [8]. Generally, the platform firm plays a leading role in the innovation process of PIE [9,10]. Accordingly, scholars started to focus on the synergic effect between platform and complementors, and pointed out that co-specialization between complementors and platform firm is the major innovative activity in PIE, but the interaction between platform firm and complementors added more complexity and uncertainty to the final innovation performance of innovation ecosystem [11]. Therefore, it is necessary to analyze the interactions between platform firm and complementors.

Although previous research has studied the interaction between complementors and platform firm [7,12–17], and revealed the importance of complementors in the process of value creation, the effects that the co-innovation process between participants in PIE have on PIE performance have rarely been known. The innovation performance of PIE refers to the effect that the members of the ecosystem bring to the whole system by their innovative behavior. The study of PIE performance in this paper is a relatively broad level, which refers to the final innovation output obtained by members of the ecosystem through interaction that includes co-opetition, co-evolution, and co-specialization [18,19]. In reality, because of the complexity and the changeability of interaction between the platform firm and complementors, relevant data is hardly available for empirical study. The interaction rises to be a prime architectural lens of the analyses and observations of PIE performance. Although the research value of the interaction has received increasing attention [20,21], the interaction does not exist in a pure form in real world. PIE usually include numerous stakeholder types. This difficulty leads us to not approaching a clear understanding of the interacting effects between the platform firm and complementors. The interacting effects, especially co-specialization, may introduce constraints and opportunities to the strategies and behaviors of participants in the ecosystem, thus influencing their performances and further shaping the evolution of the overall PIE. In evolutionary systems, the interacting members adapt and react to the co-innovation pattern that they co-create. Since the PIE of our interest is a typical complex evolutionary system, its interaction between participants should condition its evolution. So this paper simulated the collaborative innovation process between the platform firm and complementors via an improved NK model and with the simulation, it attempts to study what effects the type of complementors have on the innovation performance of the whole innovation ecosystem.

The rest parts of the paper are arranged as follows: Section 2 reviews the relevant literature. Section 3 mainly demonstrates the construction process of NK model. Section 4 is the results produced by NK model. Section 5 discusses the results and makes some suggestions for future research.

## 2 Literature review

### 2.1 Platform-based innovation ecosystem

Moore's study on the business ecosystem explained the innovation system from an ecological bionic perspective [22]. He thought that the innovation ecosystem can be regarded as an innovation system having the feature of self-organized evolution where the dynamic association of structures and functions in the system is stressed. Following his research, Some scholars focused on the regional innovation ecosystem where the different innovative groups in certain regional context are interacting with the innovative environment [23]. The research, from a

regional perspective, mainly studies the effects that the layer structures of the innovative community have on the local innovative development. Besides, the emergence of some successful innovative platforms (Intel processor, Apple smartphone, etc.) attracted extensive research interest. Gawer [1] categorized platforms as internal platforms and external platforms [1]. Internal platforms refer to the collection of the components, process, knowledge, personnel and other resources shared by products that are involved in the product development where an enterprise integrates all resources and reusable components [23]. As the accumulation of innovation resources and technology development are speeding up, the production and R&D of many components and complementary products related to a service or a product are gradually shifting to complementors. The resulting external platform can be defined as the product or service that is produced by one or more companies and capable of attracting plenty of complementary products and enriching their own values. Complementors carry out complementary innovative activities based on these external platforms. Some research on external platforms pointed out that the output of the innovation ecosystem is the service or product constituted of a few core components of lower diversity and many complementary components highly diversified [24]. While the platform firm utilizing the key advantages and updating iteratively, they collaborate with the complementors in a modularized way so as to strengthen the stability of the ecosystem and effectively stimulate the innovation efficiency of the innovation ecosystem [25]. The activities connected via the external platform through interfaces become modules that can be easily exchanged and changed without affecting the overall innovation ecosystem. This results in a architecture, usually with a relatively stable platform core and flexible connected modules [26]. In PIE, the platform along with the different complementary businesses needs to reach a collaborative and mutually beneficial strategy scheme and pursue the value co-production [8]; the participants of the platform need to realize sharing and complementing processes of their resources in forms of knowledge, technologies, assets, etc [27]. Due to the connection between interactions and value co-creation, platform research need to explore the effect of complementing processes on the generation of innovation [28]. As the important participants of an innovation ecosystem, complementors' interactions with the platform will definitely influence the innovation performance of the ecosystem.

## 2.2 Platform firm and complementors

Previous research indicates that the modularity degree of innovation output from PIE is higher and higher, and the modularized innovation come from individuals but interdependent innovative participants (platform firm and complementors) [12]. A platform firm is usually the core platform supplier of an ecosystem. The original parts provided by upstream suppliers, the component produced by the platform firm and the complementary components matching the downstream consumer market constitute the complete value chain of the innovation ecosystem [8]. The framework of PIE is led by the platform firm, which provides the platform, and the related participation rules are made by the platform firm [29,30]. Complementors are different from the suppliers. The latter are usually located in the upstream of the value chain and have a relatively stable dependency on core enterprise but lack interaction with users, while complementors are the enterprises chosen in the innovation activities of downstream participants or users and do not have fixed dependency on the platforms [31]. In terms of structure, a platform firm and different types of complementors form a "center-spokes" structure [32]. The complementarity of components can be concluded as the association where the boundary value of a variable increases as the boundary values of the other variable increase [13]. Some scholars turned to the study of relations between the platform firm and complementors, where they are usually connected in the aligned form. The aligned relations between them are usually

dominated by the platform firm, through which the ultimate customer group can decide freely to combine the product of a platform firm with which complementary product [33]. Some research indicates that any improvement of general technologies by complementors can promote the productivity and innovation capacity of downstream industries [34,35]. However, complementors' dependency on certain ecosystems increases the risk of opportunistic behavior of the dominant platform firm, while with more complementors choosing the ecosystem, such risk will decrease [36]. The framework of PIE may need more complementors that can provide customized service to the platforms. These complementors can provide complementary products that are customized or compatible with the platform firm in order to satisfy the one-way demand from the platform for complementors [1,37]. However, while the cooperation of complementors and platform firm deepening, the platform firm may lose its leading position gradually [38]. From the perspective of economic relations, Teece [39] divided the complementary assets affecting the innovation, into two types–general and special. General complementary assets can be regarded as the complementors' assets involved in the innovation process, and these assets do not need to be tailored according to the products or services of core enterprises. For example, the Android system of Google can be installed and operated on smartphones of different brands. On the contrary, the assets that have to be tailored are special complementary assets [39]. For example, iPhone Lightning Dock, the charging port can only be used in the series of smart products produced by Apple. The previous research shows that different types of complementary assets are a kind of important assets that can promote innovative entities to gain more value from innovation [40–42]. However, with the advent of many digital technologies (e.g., cloud computing, 3D printing, blockchain), the cycle of value creation and value capture is shortening [43] and the boundaries between the process and gains of innovation are ambiguous [44,45]. In addition, complementor strategies are more complex than simply trying to maximize coopetition [46]. Coopetition relationships involving different types of complementors have high levels of stress, and a major risk that these complementors have is the partner of elimination [47]. We argue that complementors, which complement to platform, can be divided into generalist and specialist. From the perspective of whether the complementors will reject others in collaboration, some research points out that decreasing the collaboration with complementors of exclusiveness or nonexcludability will benefit the platform firm [48]. From the perspective of the knowledge conversion rate, Teodoridis [49] indicates that different types of complementary participants in the innovation should make more collaborations in order to promote innovation performance [49]. But the studies mentioned above do not identify the influence of either generalist complementors or specialist complementors on the innovation performance in PIE. In order to promote innovation performance, a platform firm as the core of an innovation ecosystem needs to choose collaborative partners from both types of complementors [50]. The purpose of this paper is to study the effects of the collaborative innovation process between the platform and complementors on PIE performance.

## 3 Model and simulation

### 3.1 Basic model

NK model was originally designed to study the biological evolution [51], and was later introduced to the research of strategic management [52]. For example, with NK model, Mihm [53] studied the problem of how the complexity of a system affects its overall performance [53]. NK model has been demonstrated to be useful in simulating the innovation and the collaborative interactions between system modules by showing the effects of changes of the system modules on the overall performance of the system [54]. And also, some research based on NK model

proves the influence of technological complexity on innovation management and decision-making [55]. NK model has been popular with the strategy and organizations literatures because the methodology captures fundamental ideas of these studies, namely, that system-level results depend on the performance of multiple interacting components, therefore, successfully managing such a system requires keeping an eye on cognitive of managers and limitations of organizations [56]. Ganco [57] verified that the innovative process may be successfully represented using the NK model through the empirical approach [57]. In this paper, we try to trace the co-innovation process back to the members of PIE, and to understand the dynamic behaviours of collaborative interactions.

NK model consist of two important parameters–N and K. N represents the number of components that constitute the overall genotypes of the subjects. N can stand for the number of resources controlled by a system [58], the number of ports [59], and the overall structure of the whole system [60], which implies that the N components of the whole system all have an influence on the system. K indicates the overall degree of cross coupling and interactive degree between the components. These interactions refer mainly to the process where changes of certain factors or components in the system will cause other K factors or components to change. For instance, Fig 1(A) shows when N = 3, K = 2, the relations between components. The arrows demonstrate that the contribution of every component towards the whole system is influenced by the other two components. The value of K also reflects the complexity of the system. The value range of K distributes within (0, N-1). When K = 0, it means all of the N components have no direct link between one another, and the system has the lowest complexity. When K = N-1, the complexity of the system is at the highest degree, which means that every component of the system has links to the other all components in the system. Generally, suppose that every component has a state of 1 or 0, N components will have $2^N$ possible states. Following the NK methodology, the Parameter Ā represents the value of the states. In the improved NK model, the value do not necessarily mean the absence or presence of a given component but, rather, just two different ways in which a particular component can be configured. For example, Fig 1(B) presents, when N = 3, all the combinations of the subjects involved in the model.

Complex systems are composed of components that are interdependent in complex ways. It is necessary to to explore the non-linear process of generating creative behaviors from NK model thinking. We make the implicit assumption that the object made up of components are largely decomposable, meaning that the impact of a given component on fitness is independent of the values of other components. Each component is assumed to have its "own" fitness. The fitness component of each element is affected by the same number (K) other components.

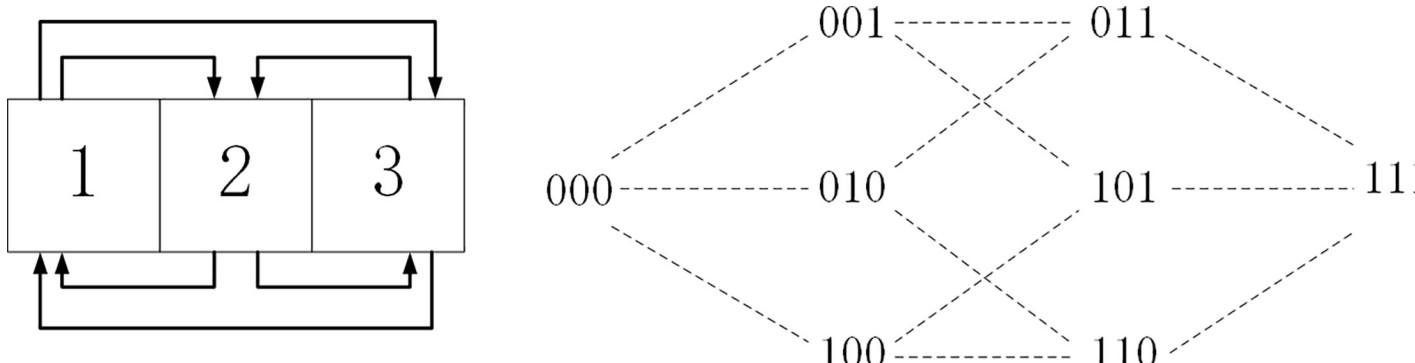

**Fig 1. The relationships between the NK model components and the N possible representations.** Note: N = 3, K = 2. (a) description of NK model relationship (b) combined form spatial distribution.

Each component affects one or more fitness components. The object composed of N components have an additive fitness structure in that the fitness contribution of each component. The overall fitness (F(n)) can be considered as the mean of each component fitness($f_i$). The value of fitness can be seen as random variables ranging within (0,1). The fitness value of the system is calculated by adding all the values of N components and then reaching the mean value of the sum according to formula (1). In the formula, F(n) represents the overall fitness, $f_i$ represents the fitness of each component to the total contribution, and the expression form of $f_i$ ($n_i$,...$n_{i+K}$) reflects that the ith component is affected by k other components.

$$F(n) = \frac{1}{N} \sum_{i=0}^{N-1} f_i(n_i, \ldots n_{i+K}) \tag{1}$$

By matching every combination form in Fig 1(B) with a fitness value, a fitness landscape with peaks and valleys will appear. The number of peaks in the fitness landscape will increase as the value of K increases as shown in Fig 2. We assume there is an agent that can produce N components. In this way I have mapped the change of N components and the movement of agent the across the landscape. When an agent has produced a program with N components, the agent appears in the fitness landscape composed of all the combination of this program. While agents have bounded rationality and can only see the adjacent programs, and every one of the programs has only one of the N components changed. Agents at the same time have also the appeal to increase their fitness values. Agents move onto an adjacent program with higher fitness value by changing the current state of any components. When agents move toward higher fitness values, we have to use the searching method to "climb the slope" in the fitness landscape [52]. When there are no bigger fitness values in the adjacent combination states, the agent will stay steady. Past research thought this position of steady state might be a local optimal value, which is a peak value of many peak values [51].

## 3.2 Model

This paper assumes that a service or product output from an innovation ecosystem is composed of some components provided by different types of complementors and other components provided by a platform firm. This paper uses A and B to represent complementor and platform firm respectively. It is worth noting that every A can represent a single complementor, but the corresponding B may only represent a department or subsidiary of the platform firm, or part of the ports that the platform firm has opened to a complementor. The combination of complementor A and platform firm B represents a complete product or service. The assumption mentioned above means that the output of the innovation ecosystem consists of several parts. For example, the hardware of smartphones (screen, battery, chip) along with the operating system and software [61], new energy automobiles and their supporting charging facilities [31] are all typical PIE parts. This paper also made the assumption that the final service or product of PIE is composed of N components including platform firm product, complementary product, and collection of related technical standards supporting the interoperability. This paper does not cover the studies of product size of complementor and platform firm, and the analysis of the dominance of innovation participants through controlling the number of components. Therefore, we assume here that both complementor A and platform firm B take up N/2 of all components and the positions of the components controlled by both sides are continuous. Hence a PIE can be denoted by <A, B>. The products and services produced by complementor A and platform firm B can be signified as A = <$n_1$,$n_2$,..., $n_{N/2}$>, B = <$n_{N/2+1}$,$n_{N/2+2}$,...,$n_N$>. The illustration of co-innovation process based on the improved NK model and the corresponding parameters is shown in Table 1.

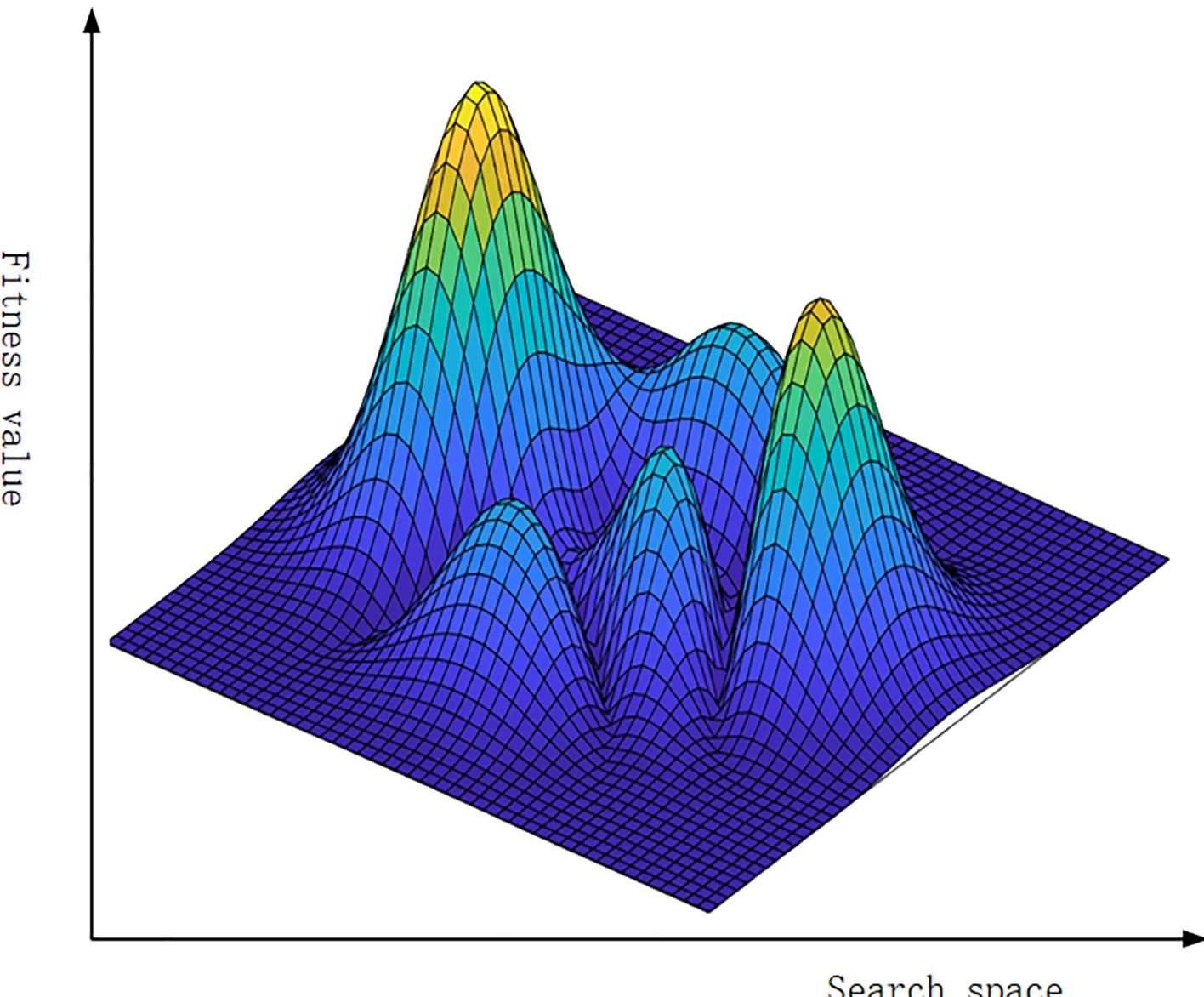

**Fig 2. Fitness landscape of the NK model.**

In this paper, we signify two kinds of different complementors (generalist and specialist) by changing the parameter configuration of NK model. Each type complementor controls the same number of components, so the features of complementors are mainly decided by the quality differences of the components. For example, a certain complete service or product consist of eight components (N = 8). The relations between different components are not independent or simply mechanical links. They are actually interdependent and coupled. For example, a database product of Oracle needs code support from Java and server with proper service configuration as hardware support. Among eight components, if complementor A controls four components ($N_A = 4$, A = $<n_1,n_2,n_3,n_4>$) and if A is a generalist complementor, then its components should have a combination freedom of high degree, which indicates a higher possibility of cooperating with more platform firms. No matter it is complementor A or platform firm B, their internal componentsare inter-dependent and interactive. Therefore, we use K to

**Table 1. The proposed model for co-innovation process based on the NK model.**

| Parameter | Classic NK model | Improved NK model |
|---|---|---|
| N | The number of parts in a system | Components that make up the products of the PIE |
| K | The number of other parts which influence each N | The number of other Components which influence each N |
| Ā | Number of alleles at each site | Possible states for each component |
| Fitness values | the average of the contribution of all parts | Innovation performance of the PIE |

represent the association degree between the groups. Suppose the association degree of complementor A is 2 ($K_A = 2$), which means every component of complementor A has the other two components related to it. As shown in Fig 3, light blue blocks and deep blue blocks represent the components contained in generalist complementor A and the related components of former components. The components contained in generalist complementor A and the related components of former ones are not embedded into the components of the corresponding platform firm. These complementors are relatively independent, and have sufficient authority to modify the components and can realize the possibility of docking with multiple platforms. If A is a specialist complementor, then its components should appear to be more dependent on the components of the platform, so the components produced by specialist complementors can be regarded as the tailored products of the platform-produced products. As to the specialist complementors, as indicated in Fig 4, light blue and deep blue blocks represent the components of the specialist complementors A and the related components of the former ones. The related components of the specialist complementors A controlled by the platform firm which works with specialist complementors A to prduce the final service or product. The deeper the cooperation between the specialist complementors A and the platform firm, the higher the coupling degree between components.

After clarifying the features of both types of complementors, this paper continues to analyze and establish NK model demonstrating the interactions between a platform firm and complementors of different types. In Figs 3 and 4, the different positions of light red blocks compared to those of deep red blocks represent the different interdependent situations between the platform firm and complementors in (a), (b), (c). According to the formerly made assumption, in the situation where $N_A = 4$, $K_A = 2$, the component number of platform firm B is 4 ($N_B = 4$,

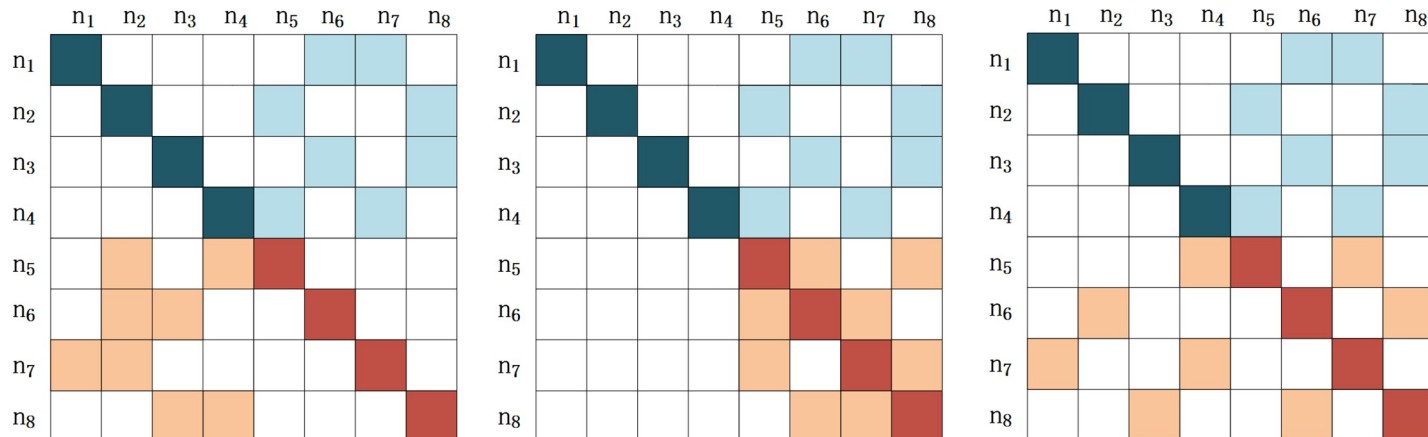

**Fig 3. Three dependent situations of specialist complementors participating in the collaborative innovation process.** Note: N = 8, K = 2. (a) dependent platform firm (b) independent platform firm (c) ambiguous dependency.

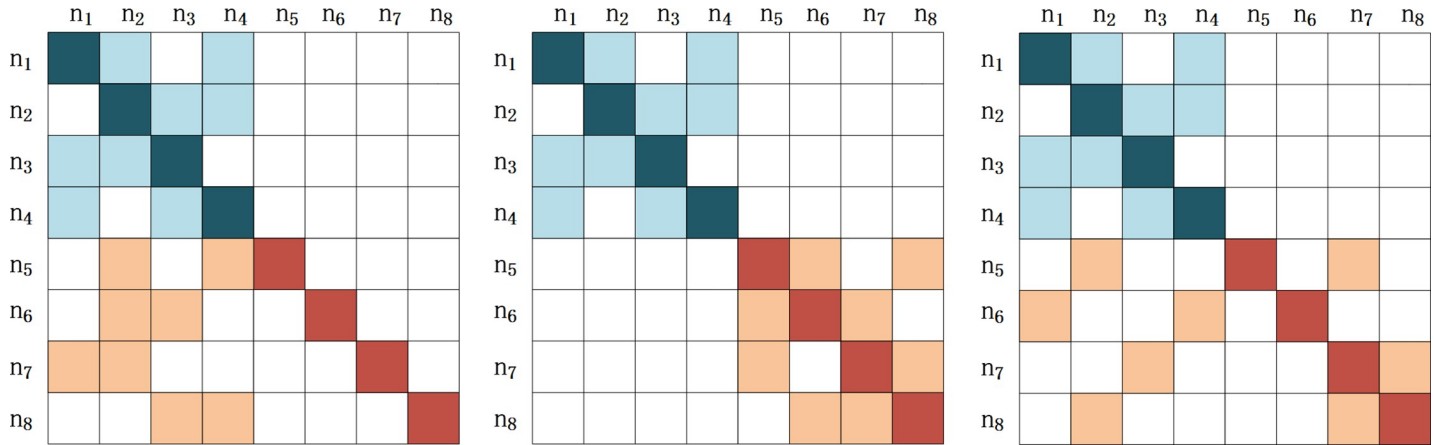

**Fig 4. Three dependent situations of generalist complementors participating in the collaborative innovation process.** Note: N = 8, K = 2. (a) dependent platform firm (b) independent platform firm (c) ambiguous dependency.

$B = <n_5,n_6,n_7,n_8>$). Likewise, in the collaborative innovation process there are other components ($K_B = 2$) related to the components of platform firm B. There are three possible situations of platform firm B interacting with complementors:

1)  The platform firm is highly dependent on the complementors, as indicated in Figs 3 and 4 (A). All of the components related to the components of platform firm B belong to the components of the complementor A.

2)  The platform firm and the complementor are independent. In other words, the platform firm does not rely on the complementor as indicated in Figs 3 and 4(B), where the components of platform firm B are not related to the components of complementor A.

3)  The relation between the platform firm and the complementor is ambiguous, as in Figs 3 and 4(C). The components related to the components of platform firm B unpredictably controlled by the platform firm or the complementor.

Regarding the combination $<A, B>$ as an agent, then the innovation process towards a service or product containing N components can be described as the slope-climbing process of the agent in the fitness landscape. The search method of NK model used in this paper is a local search. In other words, an agent (representing platform or complementor) changes the state of only one component, and through constant trial and error, it moves to the states of adjacent components in order to reach a higher fitness value. This process can be compared to the incremental innovation behaviors of the members in the innovation ecosystem. The innovator takes a local adjustment strategy rather than a disruptive change of the existing combinations of components. The overall innovation performance of the innovation ecosystem can be represented as the joint performance of the complementors and the platform firm. The overall performance is the sum of performances of A and B. According to the definitions mentioned before, the overall innovation performance equation can be represented as:

$$F_A(n) = \frac{2}{N}\sum_{i=0}^{(N-1)/2} f_i(n_i, \ldots n_{i+K/2}) \tag{2}$$

$$F_B(n) = \frac{2}{N}\sum_{i=(N+1)/2}^{N} f_i(n_i, \ldots n_{i+K/2}) \tag{3}$$

$$F(n) = F_A(n) + F_B(n) \tag{4}$$

An agent that includes a specialist complementor explores in the NK landscape, and the change in its innovation performance value includes two parts, in which the complementor provides the part to consider the impact from the platform components. In turn, when the agent of a generalist complementor explores the landscape, the innovation performance value provided by the complementary enterprise only considers the influence of its own components. For example, let us use N = 4, K = 1 for ease of notation. If an agent currently sits at 0101, it can move to 1101,0001,0111,0100, regardless of which type of complementors this agent contains, but the calculation of innovation performance is different because of the distribution of components. If the agent includes a specialist complementor, $F_A(01)$ can be represented as:

$$F_A(01) = \frac{1}{2}(f_1(n_1 n_3) + f_2(n_2 n_4)) = \frac{1}{2}(f_1(00) + f_2(11)) \tag{5}$$

If the agent includes a generalist complementor, $F_A(01)$ can be represented as:

$$F_A(01) = \frac{1}{2}(f_1(n_1 n_2) + f_2(n_2 n_1)) = \frac{1}{2}(f_1(01) + f_2(10)) \tag{6}$$

Similar to the above representations, the three situations of the platform also affect the components distribution of B, and then the innovation performance values.

## 3.3 Simulation

This paper used the Sendero simulation model developed by Padget [62] to test the effects of different types of complementors on the overall innovation performance as the association between components gets stronger (value of K increases). Suppose that N = 16, K = 2,. . .,14 (because complementors and the platform firm need to take 1/2 of the values of K and N, so K and N have to be even numbers). According to the assumption before, the values can be $N_A = N_B = N/2 = 8$, $K_A = K_B = K/2 = 1,. . .,7$. In this paper, under the formerly mentioned three dependency situations, we recorded the comparison simulation experiment on the collaborative innovation process between platform firm, and generalist complementors and specialist complementors. At the same dependency and association degree of components, the effects of the interactions between different types of complementors with the platform firm, on the overall innovation performance are observed. To ensure the accuracy of the experiment, the simulation experiment under every situation is operated for 1000 times as below:

Put 100 agents into the innovation performance landscape, as indicated in Fig 5, every agent represents the alliance of the complementors and the platform firm producing complete products. All agents are uniformly distributed in the performance landscape through improved program simulation settings. These settings jointly ensure that, the distribution network formed by the agents does not affect the simulation results. The agents by evaluating the adjacent component combination change anyone of the N/2 components to move to a better position. It is worth noting that the agents can only change the component states controlled by complementors, which are the states of $n_i$ in A = $<n_1,n_2,. . ., n_{N/2}>$. The components of platform firm B are more like the anchors of mega-vessels, which have anchoring effects in the innovation process. The innovation activities of complementors are centered around the products of the platform firm. For example, software platforms such as the Apple platform or application software ecosystem are usually the standards or ports, complementors need to update

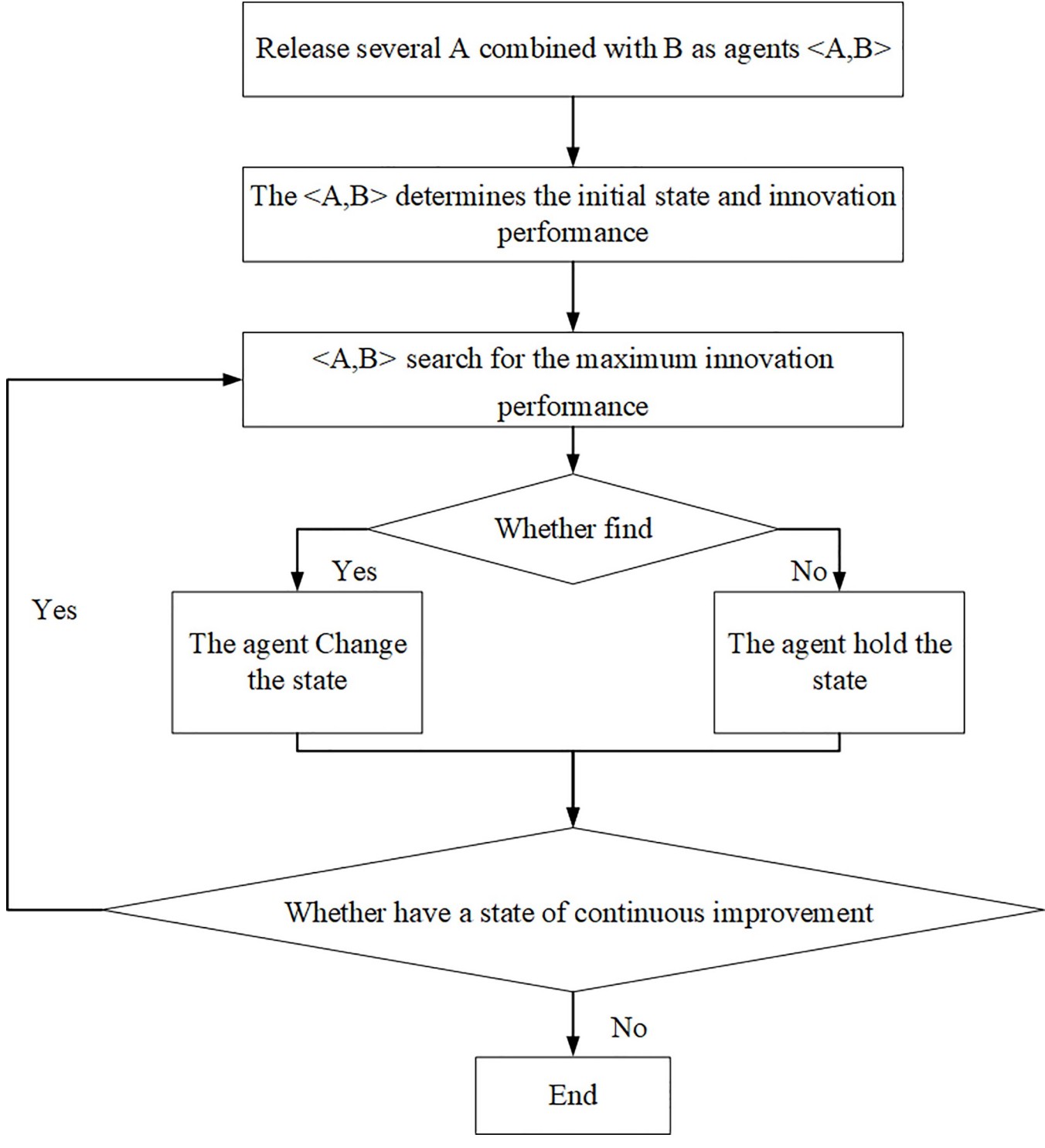

**Fig 5. Simulation process.**

and iterate their services or products in accordance with the components or docks of the platform firm. When all the 100 agents stabilized in the innovation performance landscape, which means that there is no need to further improve the components, then we can end this

simulation and record the average fitness value of the 100 agents. Thereafter, repeat the simulation process again.

## 4 Simulation results and analysis

Under the three different dependency situations, we calculated the average values of the 1000 simulation results of association degrees ($K_A$) of different components and analyzed the results.

Fig 6 shows the simulation results when the platform firm (Figs 3 and 4(A)) is dependent on the complementors. Fig 6 plots the average of co-innovation performance value for each $K_A$. At $K_A = 4$, for example, average performance value under generalist complementors co-innovation is 0.6594 (with variance .0029), and under specialist co-innovation it is 0.6345 (with variance .0028). The bars are standard errors of the mean. Generally speaking, in the situation of a dependent platform firm, as the dependency between complementary components is deepening (value of $K_A$ increases), the co-innovation performances between two types of complementors and the platform firm both tend to be an inverted U-shape. Innovation performance of generalist complementors is better than that with the specialist ones. At the intermediate association degree ($K_A = 2,3,4$), the innovation process involving generalist complementors has obvious advantages over those without the involvement. As the value of K increases, the co-innovation performance of generalist complementors decreases faster than that of specialist complementors ($K_A = 6,7$). The most typical cases are the mobile operating systems iOS (App Store) and Android (Google Play). Both platforms entail co-innovation and it is through this co-innovation that two digital platforms provide opportunities for complementors. They provide boundary resources, such as application programming interfaces,

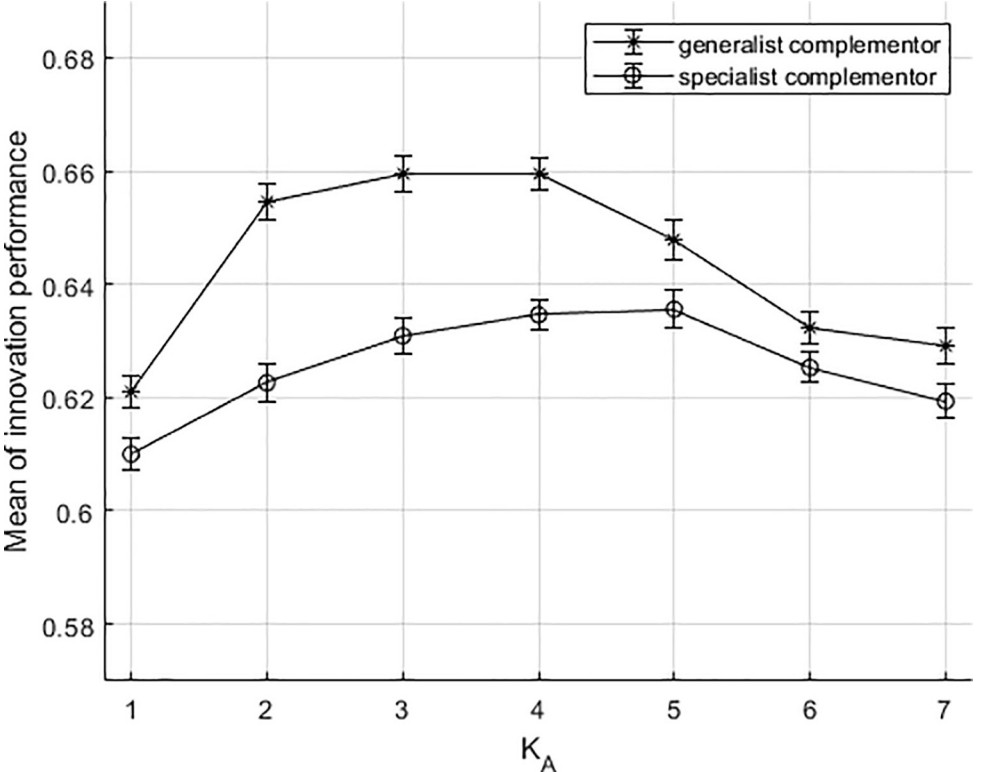

**Fig 6. The results of the simulation under dependence situation.**

software development kits to enable complementors to enhance the value of the ecosystem. These actions reflect the platforms' reliance on complementors. By their nature, innovation activity differs substantially between them as a result of differing design and architecture. The complementors collaborating with Google Play are more like generalists. The boundary resources from Google is nearly free [63]. The complementors with broad niche width offer a generic technology that can be employed to create solutions to serve other digital platforms. On the contrary, the complementors of iOS are more like specialists. The IOS platform strategy limits the number of complementors to only those that develop offerings in restricted applications, precluding complementors that replicate the same product to other digital platforms [64]. While it is difficult to compare the innovation performance of the Apple ecosystem with the Android ecosystem, the available data shows that Google's user downloads continue to be higher than Apple's. Android catch up with iOS by generic developers' skills and pooling resources. As more complex and uncertain new technologies emerge, such as 5G, the pace of innovation for both digital platforms has slowed.

In the situation of an independent platform firm, the simulation results are shown in Fig 7. Compared with the results of the dependent situation, the differences are: first, innovation performances of the collaborative innovation between both types of complementors and the platform firm are all lower than that in independent situation; second, the innovation performance of generalist complementors is still higher than that of specialist complementors; third, compared to specialist complementors, the collaborative innovation participated by generalist complementors presents noticeable advantages, and the positions move a little bit ahead compared with former situation ($K_A$ = 1,2,3). The co-innovation performances between two types of complementors and the platform firm still both tend to be an inverted U-shape, but

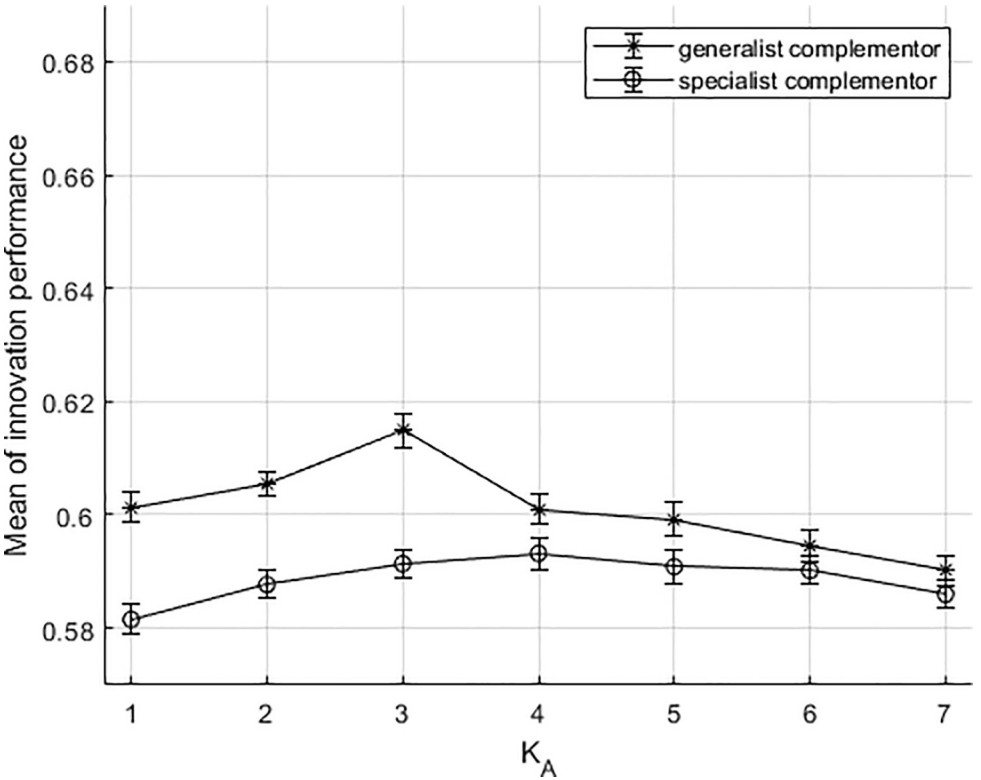

**Fig 7. The results of the simulation under independence situation.**

the curves of both have obviously decreased. In comparison, the former situation is more beneficial to the improvement of innovation performance in the innovation ecosystem. This result implies that the situation of a platform firm being dependent on the complementors is more beneficial to promote the overall innovation performance. The complementarities between electric vehicle and the charging infrastructure are rather obvious [31]. the electric vehicle ecosystems are more like the independent situation. The combination of electric vehicle enterprises and charging enterprises is more flexible. The electric vehicle charging market is a highly differentiated, in which there are different types of complementors. Since the market for infrastructure construction of electric charging stations has been opened in China, the development of electric vehicle ecosystems has been greatly affected. Many enterprises such as Beijing Energy Investment Holding, Star Charge,Huashang, Sanyou and CATL tend to be generalists in the process of cooperation with electric vehicle enterprises. These enterprises needed to adapt their production to develop compatible connector and build charging piles that match different types of electric vehicles [65]. Generalists of the electric vehicle ecosystem can achieve interconnection, as interface and standards are unified. Generalists also have the advantage of better grasping the optimal matching degree between charging piles and electric vehicles. In contrast, Tesla build and arrange specialized network of electric charging stations offering charging services for their electric vehicles. Potevio as charging enterprise is more like specialists in its partnership with Tesla. The charging piles from Potevio offer service to only electric vehicles from Tesla. Although Tesla and Potevio have improved compatibility in subsequent production, reports on the compatibility of Tesla electric vehicle charging can often be heard from the media. It has also become one of the important reasons that affect Tesla electric vehicle sales.

When there is an ambiguous dependency between the platform firm and complementors, the results of the simulation are shown in Fig 8, where the overall state of the innovation process participated by the two types of complementors shows a trend of fluctuations. The third situation is mainly used to compare the first two situations. This situation fully shows that in the real world, if the academic research does not fully abstract the platform's dependence on complementors, everything will become chaotic. In the stage of a lower association degree between components ($K_A$ = 1,2,3,4), the innovation process involving generalist complementors has again outperformed those involving the specialist ones. As the association degree increases, the innovation performances of the two types of complementors have gradually converged. Because the relations between the platform firm and the complementors cannot be identified, so the differentiation advantages of complementors are not embodied. The simulation results illustrate that the way the improved NK model is set up it can only account for a twofold distinction. Whether the dependency between the platform firm and complementors is clear is a very significant prerequisite for the simulation.

To further validate the results of the simulation experiment, we next repeat the above three scenarios by increasing the value of N. Suppose that N = 32, K = 2,. . .,30 (the value of N is close to the ceiling of our simulation experiment). According to the assumption before, the values can be $N_A$ = $N_B$ = N/2 = 16, $K_A$ = $K_B$ = K/2 = 1,. . .,15. Holding all other simulation conditions constant, the simulation experiment under every situation is also operated for 1000 times as the same simulation process. Fig 9 shows the simulation results under the three different dependency situations. The curve trends in Fig 9(A) are consistent with Fig 6. The co-innovation performances similarly tend to be an inverted U-shape. the trend of the curves in Fig 9(B) is also consistent with that in Fig 7. The results in Fig 9(C) also conform to the above analysis of the simulation results of the third situation. Regardless of how the value of N changes, the results of the simulation in this paper are similar. This shows that the value of N has little influence on the simulation of the collaborative innovation process.

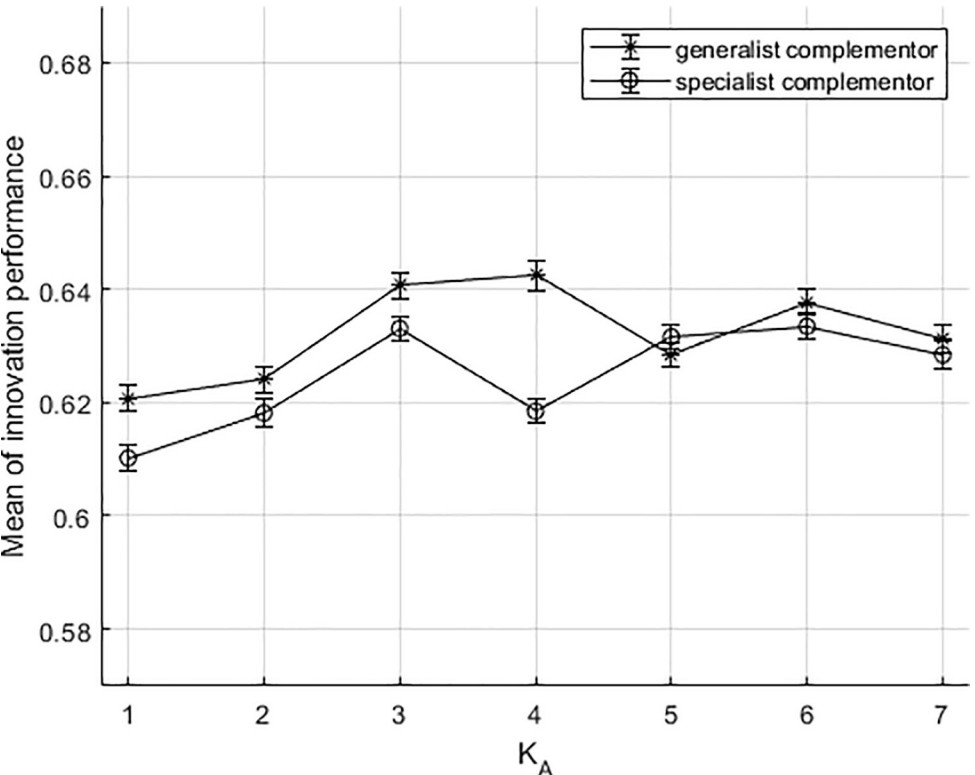

**Fig 8. The results of the simulation under ambiguous dependency situation.**

## 5 Conclusion and discussion

Under the consideration that previous research has seldom paid attention to the collaborative innovation process between the platform firm and complementors and their effects on PIE performance. Previous empirical methods can neither analyse and explain the mechanism of collaborative innovation process nor analyse the behaviours of innovative participants of PIE. This paper attempts to fill the gap by simulation. Through considering the interdependencies between platform firm and complementors, this paper make simulation experiments. The results indicate that: as the association degree between complementary components in PIE

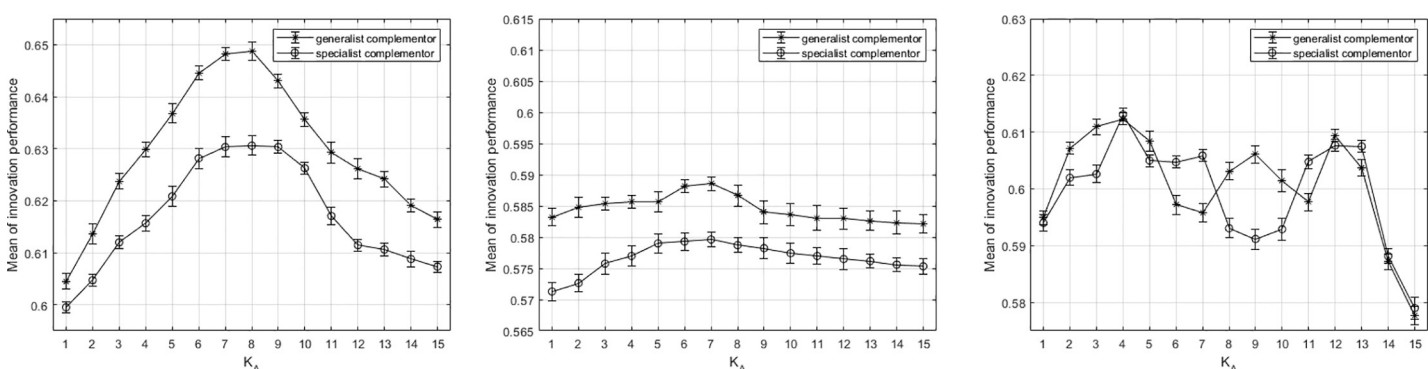

**Fig 9. The results of the simulation under three dependent situations Note: N = 32.** (a)dependent platform firm (b)independent platform firm (c)ambiguous dependency.

increases (value of K increases), generalist complementors have different performances from specialist complementors in participating in the process of the collaborative innovations. Complementors affects innovation performance of the PIE through different complementarities, as they convey distinct levels of component heterogeneity and a willingness to compete against each other. As the association degree increases, the active participation of complementors can benefit the promotion of advantages of the entire ecosystem. As is shown in case study by Baldwin [30], towards the IOS and Android smartphone ecosystems [66], the innovation performance of the innovation ecosystem is affected by the complexity of the system and the dependency between innovation entities. Compared with the simulation results of this paper, the dependence relationship between platform and complementors represents the strong support of platform to complementors. The characteristics of interdependence also become the fulcrum of complementors to participate in competition. The complexity of the innovation ecosystem also represents the scale of the innovation ecosystem. The results of this paper also show that there is an obvious critical point in the evolution of the innovation ecosystem, and this critical point can only be clearly revealed when the members of the innovation ecosystem exist as a pure type and have an explicit dependence relationship.

Parameter N of the model represents the number of modules that make up the final product or service of PIE. Parameter K of the model can be understood as a hypothesis that there is an association between the modules of the platform firm and complementors. Simulation results also have indicated the strong association between the modules of both sides is beneficial to the promotion of the innovation performance in the innovation ecosystem. Through the observation of the different situations of the platform firm and complementors, it is found out that all the operations between generalist complementors and the platform firm are better than those between specialist complementors and the platform firm. One important reason behind this phenomenon might be that generalist complementors stress more on the compatible functions of the platform ecosystem [67]. Generalist complementors are usually able to share resources with many platforms, and the success of a platform is usually related to its ability to attract more generalist complementors to participate in innovations [68].

The simulation results also show that in the situations of identified dependency between the platform firm and the complementors, as the association degree between complementary components changes, the process of collaborative innovation participated by both generalist and specialist complementors changes an inverted U-shape. With the change of innovation complexity, different innovation participants of PIE may show differences in the acquisition of innovation resources, driving mode and R&D focus. generalist complementors and specialist complementors are actually engaged in different types of innovation. We try to find the "critical point" of two kinds of complementors. The critical point is that when the complexity of innovation ecosystem reaches a certain degree, the promotion effect of complexity on system innovation disappears, and the deepening of innovation ecosystem complexity inhibits system innovation. The inner association degree of the complementary components also represents the influence levels of the product complexity on the innovation performance in the innovation ecosystem. Therefore, the more complicated a product is the more are its functions and features [69]. However, when the product complexity reaches a certain level, the value of the product or service may decrease [70]. What is unique about this paper is: first, it focuses on the process of co-innovation between the platform firm and complementors. Ignoring such a process will lead to obvious mistakes in the theoretical research of innovation ecosystem. Second, according to the different dependency between platform and the different types of complementors, this paper depict the behaviours of them that participate in co-specialization. By distinguishing the influence of the types of complementors on the innovation performance of the innovation ecosystem, this paper deepens the research of collaborative innovation, which is

helpful to reveal the black box of interaction between members of PIE. Third, The collaborative innovation process simulated in this paper takes place outside the participants of PIE and the actual situation is often unable to peel off the internal business behavior of the participants (such as profit margin, return on capital, return on equity, etc.) to fairly compare the simple collaborative innovation behaviors. The simulation in this paper can help us to better understand the process of co-innovation of PIE in future research. The limitation of this paper is as follows. The factors influencing collaborative innovation are complex. Innovation behaviors of participants in different PIEs cannot be generalized. Innovation in either of them could be distinct due to resources availability. Meanwhile, the simulation in this paper is closer to the actual situation of the emerging information industry, and therefore inapplicable in other industries. Future studies should fill this gap. In future, simulation for innovation ecosystem should give cognizance to the different types of industry.

## Supporting information

**S1 Data. Data of simulation.**
(ZIP)

## Author Contributions

**Conceptualization:** Fangcheng Tang.

**Supervision:** Fangcheng Tang.

**Writing – original draft:** Zeqiang Qian.

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
