## [Decision Letter · Decision Letter 0]

23 Mar 2020

PONE-D-19-35558

Leveraging Interdependencies among Platform and Complementors in Innovation Ecosystem

PLOS ONE

Dear Dr. Qian,

Thank you for submitting your manuscript to PLOS ONE. After careful consideration, we feel that it has merit but does not fully meet PLOS ONE’s publication criteria as it currently stands. Therefore, we invite you to submit a revised version of the manuscript that addresses the points raised during the review process.

Though Reviewer 1 rejected PONE-D-19-35558, the reviewer provided many valuable and constructive comments. Considering three reviewers’ useful comments and the interesting topic of the manuscript, I would like to give you a chance to revise your manuscript during the special period. The revised manuscript will undergo the next round of review by the same reviewers.

We would appreciate receiving your revised manuscript by May 07 2020 11:59PM. To enhance the reproducibility of your results, we recommend that if applicable you deposit your laboratory protocols in protocols.io, where a protocol can be assigned its own identifier (DOI) such that it can be cited independently in the future. For instructions see: http://journals.plos.org/plosone/s/submission-guidelines#loc-laboratory-protocols

We look forward to receiving your revised manuscript.

Kind regards,

Baogui Xin, Ph.D.

Academic Editor

PLOS ONE

Journal Requirements:

Reviewers' comments:

Reviewer's Responses to Questions

**Comments to the Author**

1. Is the manuscript technically sound, and do the data support the conclusions?

Reviewer #1: Partly

Reviewer #2: Partly

Reviewer #3: Yes

2. Has the statistical analysis been performed appropriately and rigorously? 

Reviewer #1: I Don't Know

Reviewer #2: Yes

Reviewer #3: I Don't Know

3. Have the authors made all data underlying the findings in their manuscript fully available?

Reviewer #1: No

Reviewer #2: Yes

Reviewer #3: Yes

4. Is the manuscript presented in an intelligible fashion and written in standard English?

Reviewer #1: Yes

Reviewer #2: Yes

Reviewer #3: No

5. Review Comments to the Author

Reviewer #1: The paper studies the effects of collaborative innovation process in a Platform-based Innovation Ecosystem (PIE) using simulation. The topic is interesting, however the contribution of the work is not sufficient to justify publication in PLOS ONE, while several parts of the submitted paper are not clearly presented. My major comments are the following:

1. The paper focuses only on the co-specialization process in innovations ecosystems. However, as noted by several authors, interactions should consider the dynamically intertwined processes of co-opetition, co-evolution, and co-specialization (see for example Carayannis, 2001).

2. A simulation-based approach, without any empirical evidence, is not sufficient to study the aforementioned problem. Current work looks more like an academic exercise.

3. The adopted model, as given in section 3, is not clearly presented. For example, every variable in equations should be rigorously defined, assumptions should be better justified, simulation parameters should be clearly defined and justified, etc. For example, the authors note that they put 100 agents randomly into the innovation performance landscape. But what “randomly” means? Do they mean "uniformly at random"? If yes, what parameters have been chosen for the uniform distribution? Why?

4. Several results are poorly discussed and justified. For example, the inverted U-shape relation is an important finding. The authors simply note this, but they do not give a suitable justification and explanation of this finding (the only justification of results can be found in lines 402-404).

5. Moreover, some results seem self-evident (e.g., “…the innovation performance of complementors in PIE is affected by the complexity of the system and the dependency between innovation entities…”).

6. I cannot see any significant valuable managerial implication in section 5.

References

Carayannis E.G. (2001). The strategic management of technological learning: Learning to learn and learning to learn how-to-learn as drivers of strategic choice and firm performance in global, technology-driven markets. Boca Raton, FL: CRC Press.

Reviewer #2: This paper will help us to understand well the relationship of complementors and platform firms. It is important to improve innovation performance in innovation ecosystem. I think this topic is meaningful ,and for this reason I applaud the efforts of the authors for this topic. But I think it can be improved in the following aspects.

Major issues:

1. Fig3(a)-3(c) and Fig4(a) -4(c) are not well explained. Three dependent situations of specialist complementors and generalist complementors participating in the collaborative innovation process are vague.

2.. In the simulation, you should set several groups of N to get more reliable conclusions. If N is only set to be equal to 16, the conclusions may be subjective.

Minor issues:

1. Some words are misspelled.

Reviewer #3: 1. PIE performance is a key construct in this paper. But the authors fail to address how to evaluate PIE performance and existing research about it should also be included.

2. The authors should clarify why NK model is suitable for this research.

3. More latest research literature should be included in the paper.

4. The authors should revise the language to improve readability. For example, Line 20-24 are difficult to follow.

6. PLOS authors have the option to publish the peer review history of their article (what does this mean?). If published, this will include your full peer review and any attached files.

Reviewer #1: No

Reviewer #2: No

Reviewer #3: No

---

## [Author Response · Author response to Decision Letter 0]

27 May 2020

Dear Editor and Reviewers,

Thank you so much for your comments and valuable advices. After reading and discussing your comments and suggestions, we carefully revised our manuscript. Some details and contents are supplemented in the revised manuscript. In the following, we try our best to clearly explain our revisions point to point.

Best regards,

Zeqiang Qian (corresponding author)

School of Economics and Management, Beijing University of Chemical Technology

Reviewer # 1

The paper studies the effects of collaborative innovation process in a Platform-based Innovation Ecosystem (PIE) using simulation. The topic is interesting, however the contribution of the work is not sufficient to justify publication in PLOS ONE, while several parts of the submitted paper are not clearly presented. My major comments are the following:

Comment 1: The paper focuses only on the co-specialization process in innovations ecosystems. However, as noted by several authors, interactions should consider the dynamically intertwined processes of co-opetition, co-evolution, and co-specialization (see for example Carayannis, 2001).

Author response: Thank you very much for your valuable and positive comments and they are very helpful to prove the quality of our manuscript. We have revised expression according to your comments. As the reviewer’s comments, the process of interaction between the participants involved co-opetition, co-evolution, and co-specialization. It was difficult to separate the process into a single aspect. While focusing on the co-specialization process in innovation ecosystems, this paper also shows the co-opetition between platform and complementors through simulation program, and the ultimate promotion of innovation performance also indicates the co-evolution of members of PIE. The adjustment on this part is mainly reflected in the section 1. The reference provided by the reviewer are important for us to improve the quality of this paper, we have added relevant studies of the reference to our paper. Thank you so much for your comments.

Comment 2: A simulation-based approach, without any empirical evidence, is not sufficient to study the aforementioned problem. Current work looks more like an academic exercise.

Author response: Thank you very much for your valuable and positive comments and they are very helpful to prove the quality of our manuscript. The main subjects of this paper are participants in the innovation ecosystem, including generalist complementor, specialist complementor and platform firm. It is difficult to study the innovation performance of the innovation ecosystem influenced by the interaction between participants through empirical methods. Innovation behaviors of complementors is often unpredictable. The quantification of innovation performance has long been controversial. In addition, the main subjects of this paper does not exist in a pure form in real world ecosystems. But the types of complementary firms do have an impact on ecosystem innovation performance. As we've already shown in the new version, using NK model on the research of innovation performance can yield desirable results. Thank you so much for your comments.

Comment 3: The adopted model, as given in section 3, is not clearly presented. For example, every variable in equations should be rigorously defined, assumptions should be better justified, simulation parameters should be clearly defined and justified, etc. For example, the authors note that they put 100 agents randomly into the innovation performance landscape. But what “randomly” means? Do they mean "uniformly at random"? If yes, what parameters have been chosen for the uniform distribution? Why?

Author response: Thank you very much for your suggestions. In section 3 we've added a table to illustrate the parameters. The Sendero that we used to build our NK model can well encapsulate the parameter properties of the NK model, so that we only need to focus on the relevant parts that we need to modify, having to focus on reconstructing the whole model. The agents random distribution problem mentioned in the comment is set internally in this program. Each run of a simulation program ensures that agents are uniformly randomly distributed in the fitness landscape. Thank you for your helpful comments. ________________________________________

Comment 4: Several results are poorly discussed and justified. For example, the inverted U-shape relation is an important finding. The authors simply note this, but they do not give a suitable justification and explanation of this finding (the only justification of results can be found in lines 402-404).

Author response: Thank you so much for your constructive comments, which are very helpful for improving the quality of our manuscript. We have carefully analyzed your comments and made corresponding revisions. We tried to give a suitable justification and explanation of this finding. Please see the new section 5. Thank you so much for reviewing our manuscript and providing us valuable comments.

Comment 5: Moreover, some results seem self-evident (e.g., “…the innovation performance of complementors in PIE is affected by the complexity of the system and the dependency between innovation entities…”).

Author response: Thank you so much for your constructive comments, which are very helpful for improving the quality of our manuscript. We have carefully analyzed your comments and made corresponding revisions. Please see Line 501-512. Thank you so much for reviewing our manuscript and providing us valuable comments. Please see

Comment 6: I cannot see any significant valuable managerial implication in section 5.

Author response: Thank you so much for your constructive comments, which are very helpful for improving the quality of our manuscript. We have carefully analyzed your comments and made corresponding revisions. We have made a major adjustment to section 5. Thank you so much for reviewing our manuscript and providing us valuable comments. 

Reviewer # 2

Authors proposed a novel multiple attribute decision-making method based on Schweizer-Sklar t-norm and t-conorm with q-rung dual hesitant fuzzy information. I think the paper is well written and its structure is well organized, and the study is innovative. So, I think this paper can be published in this journal by minor revisions.

Comment 1: 1. Fig3(a)-3(c) and Fig4(a) -4(c) are not well explained. Three dependent situations of specialist complementors and generalist complementors participating in the collaborative innovation process are vague.

Author response: Thank you very much for your comments, which are helpful for improving the quality of our manuscript. We have adjusted part of section 3 to reflect your comments. Please see Line 392-406. Thank you so much for your comments. 

Comment 2: In the simulation, you should set several groups of N to get more reliable conclusions. If N is only set to be equal to 16, the conclusions may be subjective.

Author response: Thank you very much for your reminder. The NK model is NP-complete for K>1.When N=16, there are already 65536locations in the fitness landscape. The complexity of the model is sufficient to describe the innovation ecosystem in this paper. In addition, we conducted several sets of experiments N take other values and all obtained similar curve trends. We chose only one set of data to explain our topic from a primary and secondary perspective. Thank you so much for reviewing our manuscript.

Comment 3: Some words are misspelled.

Author response: Thank you so much for your constructive comments We have carefully checked the whole paper and revised some expressions errors in the revised manuscript. In addition, we have carefully checked the manuscript and tried our best to correct typos and linguistic errors. Thank you so much for reviewing our manuscript and providing us valuable comments.

Reviewer # 3

Comment 1: PIE performance is a key construct in this paper. But the authors fail to address how to evaluate PIE performance and existing research about it should also be included.

Author response: Thank you very much for your suggestions. according to your comments we have made corresponding adjustments in section 1 and 2 of the research. In this paper, the process of collaborative innovation is regarded as a search process, trial and error search process in bounded space. This process is the reorganization and development of the existing things, the change and promotion of innovation performance is an important driving force in this process, the score of each position represents the innovation output efficiency of this position component combination. The existing research about it should is included in this paper. Thank you for reviewing our manuscript. 

Comment 2: The authors should clarify why NK model is suitable for this research.

Author response: Thank you very much for your suggestions. We have added some explanations to explain why the NK model is suitable for this research. Line 229-237 are the main parts that we added to illustrate this point. Thank you for reviewing our manuscript.

Comment 3: More latest research literature should be included in the paper.

Author response: Thank you very much for your suggestions. More latest research literature has been added to improve the Introduction and Section 2. Thank you for reviewing our manuscript. 

Comment 4: The authors should revise the language to improve readability. For example, Line 20-24 are difficult to follow.

Author response: Thank you very much for your suggestions. We have carefully checked the whole paper and revised some expressions errors in the revised manuscript. Line 20-23 has been modified to make it easier to understand. Thank you for reviewing our manuscript.

---

## [Decision Letter · Decision Letter 1]

17 Jun 2020

PONE-D-19-35558R1

Leveraging Interdependencies among Platform and Complementors in Innovation Ecosystem

PLOS ONE

Dear Dr. Qian,

Thank you for submitting your manuscript to PLOS ONE. After careful consideration, we feel that it has merit but does not fully meet PLOS ONE’s publication criteria as it currently stands. Therefore, we invite you to submit a revised version of the manuscript that addresses the points raised during the review process.

Though Reviewer 1 rejected PONE-D-19-35558R1, the reviewer provided many valuable and constructive comments. Considering three reviewers’ useful comments and the interesting topic of the manuscript, I would like to give you a chance to revise your manuscript during the special period. The revised manuscript will undergo the next round of review by the same reviewers.

We look forward to receiving your revised manuscript.

Kind regards,

Baogui Xin, Ph.D.

Academic Editor

PLOS ONE

Reviewers' comments:

Reviewer's Responses to Questions

**Comments to the Author**

1. If the authors have adequately addressed your comments raised in a previous round of review and you feel that this manuscript is now acceptable for publication, you may indicate that here to bypass the “Comments to the Author” section, enter your conflict of interest statement in the “Confidential to Editor” section, and submit your "Accept" recommendation.

Reviewer #1: (No Response)

Reviewer #2: (No Response)

Reviewer #3: All comments have been addressed

2. Is the manuscript technically sound, and do the data support the conclusions?

Reviewer #1: Partly

Reviewer #2: Partly

Reviewer #3: Yes

3. Has the statistical analysis been performed appropriately and rigorously? 

Reviewer #1: I Don't Know

Reviewer #2: Yes

Reviewer #3: Yes

4. Have the authors made all data underlying the findings in their manuscript fully available?

Reviewer #1: No

Reviewer #2: Yes

Reviewer #3: Yes

5. Is the manuscript presented in an intelligible fashion and written in standard English?

Reviewer #1: Yes

Reviewer #2: Yes

Reviewer #3: No

6. Review Comments to the Author

Reviewer #1: The revised version of the paper has considered some of my previous comments, however my major concerns have not appropriately addressed:

1. In my previous report I asked for empirical evidence about the presented simulation-based approach. The authors, in their response, note that this difficult and innovation behavior is often unpredictable. I can understand this difficulty, I can understand that it is not possible to have extensive empirical data, but at least some empirical evidence should be given. Otherwise, this is an excellent academic exercise, but with little usefulness.

2. As noted in my previous report, the model should be clearly presented (parameters, variables, equations, etc.). This is still unclear in the revised text. See for example the very first equation: F is the fitness of the system, but fi, ni, n have not been defined.

3. Similarly, the assumptions of the simulation model seem arbitrary, I cannot see any justification in the revised manuscript, although I asked for this in my previous report.

Reviewer #2: The manuscript is much better. But I think there are some problems. As you said, when N=16, there are already 65536 locations in the fitness landscape, the complexity of the model is sufficient to describe the innovation ecosystem in this paper. I think you should give a sufficient reason why do not you choose N=14 or N=18. You should have more evidence why N=16 is the best choise. If you did other experients, plesse list them.

Reviewer #3: in general, the author has made much modificaiton. But there are still some minor issues that need to be improved.

1. There are some long sentences which maybe hard to understand. I suggest that the author should shorten these long sentences. e.g. Line 66-68.

2. Line 12, the word "However" can be changed with "Although". In Line 118," in level of" can be change with" in forms of ".

3. The method of this manuscript is well described.

4. The research gap is not clear enough.

7. PLOS authors have the option to publish the peer review history of their article (what does this mean?). If published, this will include your full peer review and any attached files.

Reviewer #1: No

Reviewer #2: No

Reviewer #3: No

---

## [Author Response · Author response to Decision Letter 1]

31 Jul 2020

Dear Editor and Reviewers,

Thank you so much for your comments and valuable advices. After a second reading and discussing your comments and suggestions, we carefully revised our manuscript. Some details and contents are supplemented in the revised manuscript. In the following, we try our best to clearly explain our revisions point to point.

Best regards,

Zeqiang Qian (corresponding author)

School of Economics and Management, Beijing University of Chemical Technology

Reviewer # 1

The revised version of the paper has considered some of my previous comments, however my major concerns have not appropriately addressed:

Comment 1: In my previous report I asked for empirical evidence about the presented simulation-based approach. The authors, in their response, note that this difficult and innovation behavior is often unpredictable. I can understand this difficulty, I can understand that it is not possible to have extensive empirical data, but at least some empirical evidence should be given. Otherwise, this is an excellent academic exercise, but with little usefulness.

Author response: Thank you very much for your valuable and positive comments and they are very helpful to prove the quality of our manuscript. We have revised expression according to your comments. We added two study cases as empirical evidence. The adjustment on this part is mainly reflected in the section 4. Thank you so much for your comments.

Comment 2: As noted in my previous report, the model should be clearly presented (parameters, variables, equations, etc.). This is still unclear in the revised text. See for example the very first equation: F is the fitness of the system, but fi, ni, n have not been defined.

Author response: Thank you very much for your valuable and positive comments and they are very helpful to prove the quality of our manuscript. The adjustment on this part is mainly reflected in the section 3. Please see Line 249-264. Thank you so much for your comments.

Comment 3: Similarly, the assumptions of the simulation model seem arbitrary, I cannot see any justification in the revised manuscript, although I asked for this in my previous report.

Author response: Thank you very much for your suggestions. In section 3 we've adjusted assumptions. We also We rethought your previous suggestion. It is very helpful to us. Please see Line 403-407. Thank you for your helpful comments. ________________________________________

Reviewer # 2

Comment : The manuscript is much better. But I think there are some problems. As you said, when N=16, there are already 65536 locations in the fitness landscape, the complexity of the model is sufficient to describe the innovation ecosystem in this paper. I think you should give a sufficient reason why do not you choose N=14 or N=18. You should have more evidence why N=16 is the best choise. If you did other experients, plesse list them.

Author response: Thank you very much for your comments, which are helpful for improving the quality of our manuscript. The aim of our study is to employ the NK model of complex evolving systems to analyze/explore the collaborative innovation processes of the innovation ecosystem. According to previous studies(see for example Baumann, 2019), the NK model is a stochastic combinatorial optimization model with two parameters (N, K) for studying innovation. In our study, N is the components of the innovation ecosystem. Parameters N represent the size of the simulation object. Parameter N is a basic parameter. We need to focus on parameter K and the distribution of K. To sum up, considering the scale of the simulation experiment and the length of the paper, we set the parameter N=16.Thank you so much for your comments. 

Reference:

Baumann O, Schmidt J, Stieglitz N. Effective search in rugged performance landscapes: A review and outlook. Journal of Management. 2019 Jan;45(1):285-318.

Reviewer # 3

Comment 1: There are some long sentences which maybe hard to understand. I suggest that the author should shorten these long sentences. e.g. Line 66-68. 

Author response: Thank you very much for your suggestions. We have adjusted the parts. Thank you for reviewing our manuscript. 

Comment 2: Line 12, the word "However" can be changed with "Although". In Line 118," in level of" can be change with" in forms of ".

Author response: Thank you very much for your suggestions. We have adjusted the two parts. Thank you for reviewing our manuscript.

Comment 3: The method of this manuscript is well described.

Author response: Thank you very much for your recognition. This greatly encouraged us to continue our academic research. Thank you for reviewing our manuscript. 

Comment 4: The research gap is not clear enough.

Author response: Thank you very much for your suggestions. The adjustment on this part is mainly reflected in the section 2 and section 4. The references have also been adjusted. Thank you for reviewing our manuscript.

---

## [Decision Letter · Decision Letter 2]

13 Aug 2020

PONE-D-19-35558R2

Leveraging Interdependencies among Platform and Complementors in Innovation Ecosystem

PLOS ONE

Dear Dr. Qian,

Thank you for submitting your manuscript to PLOS ONE. After careful consideration, we feel that it has merit but does not fully meet PLOS ONE’s publication criteria as it currently stands. Therefore, we invite you to submit a revised version of the manuscript that addresses the points raised during the review process.

I recommend that it should be revised taking into account the changes requested by the reviewers. I would like to give you the last chance to revise your manuscript.

We look forward to receiving your revised manuscript.

Kind regards,

Baogui Xin, Ph.D.

Academic Editor

PLOS ONE

Reviewers' comments:

Reviewer's Responses to Questions

**Comments to the Author**

1. If the authors have adequately addressed your comments raised in a previous round of review and you feel that this manuscript is now acceptable for publication, you may indicate that here to bypass the “Comments to the Author” section, enter your conflict of interest statement in the “Confidential to Editor” section, and submit your "Accept" recommendation.

Reviewer #2: (No Response)

Reviewer #3: All comments have been addressed

2. Is the manuscript technically sound, and do the data support the conclusions?

Reviewer #2: Partly

Reviewer #3: Yes

3. Has the statistical analysis been performed appropriately and rigorously? 

Reviewer #2: I Don't Know

Reviewer #3: I Don't Know

4. Have the authors made all data underlying the findings in their manuscript fully available?

Reviewer #2: Yes

Reviewer #3: Yes

5. Is the manuscript presented in an intelligible fashion and written in standard English?

Reviewer #2: Yes

Reviewer #3: Yes

6. Review Comments to the Author

Reviewer #2: I can not see any revisions about my previous comments in this manuscript.You do not give the sufficient reason that you choose N=16. The choice of N is arbitrary. In NK model 0<=K<=N-1, if you do not choose appropriate N, K may be not appropriate. Thus, your simulation and conclusions may be wrong. You should give sufficient envidences about the choice of N.

Reviewer #3: This paper uses improved NK model to simulate innovation process between platform firms and completmentors. This is a good method and the autors described the method very clearly. But I find some sentences difficult to understand; for example, LIne 12-15, lINE28-29. Please rewrite these sentences.

7. PLOS authors have the option to publish the peer review history of their article (what does this mean?). If published, this will include your full peer review and any attached files.

Reviewer #2: No

Reviewer #3: No

---

## [Author Response · Author response to Decision Letter 2]

15 Sep 2020

To: PLOS ONE Editor

Re: Response to reviewers

Dear Editor and Reviewers,

Thank you so much for your comments and valuable advices. After reading and discussing your comments and suggestions, we carefully revised our manuscript. Some details and contents are supplemented in the revised manuscript. In the following, we try our best to clearly explain our revisions point to point.

Best regards,

Zeqiang Qian (corresponding author)

School of Economics and Management, Beijing University of Chemical Technology 

Reviewer # 2

Comment : I can not see any revisions about my previous comments in this manuscript.You do not give the sufficient reason that you choose N=16. The choice of N is arbitrary. In NK model 0<=K<=N-1, if you do not choose appropriate N, K may be not appropriate. Thus, your simulation and conclusions may be wrong. You should give sufficient envidences about the choice of N.

Author response: Thank you very much for your comments, which are helpful for improving the quality of our manuscript. Our simulation results do need to avoid subjective. under our existing simulation experiment environment, we set the maximum of N and rerun the simulation program. The adjustment on this part is mainly reflected in the section 4. Adjustments based on your comments do lead to get more reliable conclusions. Thank you so much for your comments. 

Reviewer # 3

Comment : This paper uses improved NK model to simulate innovation process between platform firms and completmentors. This is a good method and the autors described the method very clearly. But I find some sentences difficult to understand; for example, LINE 12-15, LINE 28-29. Please rewrite these sentences.

Author response: Thank you very much for your suggestions. We have adjusted the two parts. Thank you for reviewing our manuscript. 

Author response: Thank you very much for your suggestions. The adjustment on this part is mainly reflected in the section 2 and section 4. The references have also been adjusted. Thank you for reviewing our manuscript.

---

## [Editor Report · Decision Letter 3]

17 Sep 2020

Leveraging Interdependencies among Platform and Complementors in Innovation Ecosystem

PONE-D-19-35558R3

Dear Dr. Qian,

We’re pleased to inform you that your manuscript has been judged scientifically suitable for publication and will be formally accepted for publication once it meets all outstanding technical requirements.

Kind regards,

Baogui Xin, Ph.D.

Academic Editor

PLOS ONE
---

## [Editor Report · Acceptance letter]

22 Sep 2020

PONE-D-19-35558R3 

Leveraging Interdependencies among Platform and Complementors in Innovation Ecosystem 

Dear Dr. Qian:

I'm pleased to inform you that your manuscript has been deemed suitable for publication in PLOS ONE. Congratulations! Your manuscript is now with our production department. 

Kind regards, 

on behalf of

Professor Baogui Xin 

Academic Editor

PLOS ONE